# An Integrated Patient-Centred Medical Home (PCMH) Care Model Reduces Prospective Healthcare Utilisation for Community-Dwelling Older Adults with Complex Needs: A Matched Observational Study in Singapore

**DOI:** 10.3390/ijerph20196848

**Published:** 2023-09-27

**Authors:** Grace Sum, Silvia Yu Hui Sim, Junxing Chay, Soon Hoe Ho, Mimaika Luluina Ginting, Zoe Zon Be Lim, Joanne Yoong, Chek Hooi Wong

**Affiliations:** 1Geriatric Education and Research Institute, Singapore 768024, Singapore; sim.silvia@gmail.com (S.Y.H.S.); hosoonhoe@gmail.com (S.H.H.); ginting.mimaika.luluina@geri.com.sg (M.L.G.); joanne@rforimpact.com (J.Y.); 2Lien Centre for Palliative Care, Duke-NUS Medical School, Singapore 169857, Singapore; junxing.chay@duke-nus.edu.sg; 3Research for Impact, Singapore 159964, Singapore; 4Health Services and Systems Research, Duke-NUS Medical School, Singapore 169857, Singapore

**Keywords:** aged, integrated care, person-centred care, care coordination, healthcare utilisation, complex needs, community-dwelling, multimorbidity, COVID-19, quasi-experimental

## Abstract

The global ageing population is associated with increased health service use. The PCMH care model integrates primary care and home-based care management to deliver comprehensive and personalised healthcare to community-dwelling older adults with bio-psycho-social needs. We examined if an integrated PCMH reduced healthcare utilisation burden of older persons in Singapore. We compared the healthcare utilisation between the intervention group and coarsened exact matched controls for a follow-up of 15 months. Baseline matching covariates included socio-demographics, health status, and past healthcare use. We accounted for COVID-19 social distancing effects on health-seeking behaviour. The intervention group consisted of 165 older adults with complex needs. We analysed national administrative healthcare utilisation data from 2017 to 2020. We applied multivariable zero-inflated regression modelling and presented findings stratified by high (CCI ≥ 5) and low disease burden (CCI < 5). Compared to controls, there were significant reductions in emergency department (β = −0.85; 95%CI = −1.55 to −0.14) and primary care visits (β = −1.70; 95%CI = −2.17 to −1.22) and a decrease in specialist outpatient visits (β = −0.29; 95%CI = −0.64 to 0.07) in the 3-month period immediately after one-year enrolment. The number of acute hospitalisations remained stable. Compared to controls, the intervention group with high and low comorbidity burden had significant decreases in primary care use, while only those with lower comorbidity burden had significant reductions in utilisation of other service types. An integrated PCMH appears beneficial in reducing healthcare utilisation for older persons with complex needs after 1 year in the programme. Future research can explore longer-term utilisation and scalability of the care model.

## 1. Introduction

The global ageing population has an increased healthcare utilisation burden, particularly for older adults with complex bio-psycho-social needs [1,2]. This rise in health service use also has implications on the health system and wider society [3,4,5]. Integrated care models aim to address the high healthcare utilisation of older adults with complex needs [6,7]. One model of care is the integrated patient-centred medical home (PCMH), a relatively new care model recommended in primary care settings mainly in the United States (US) and Australia [8,9,10]. It is based on the concept of person-centred care, which aims to foster trust and relationships among patients, their families, and healthcare staff, involve patients in decision making about their health, and develop individualised care plans by multidisciplinary teams [11]. Implementation of an integrated PCMH is differentiated from usual primary care by incorporating team-based care, shared care among healthcare professionals, and an empanelment strategy [12]. The literature supports the need for complex needs patients to receive person-centred and integrated care for better outcomes [13].

Current literature has reported mixed findings on the effects of the PCMH model of care on health service use. This is likely due to the implementation of a multicomponent intervention in heterogeneous settings and populations and examination of outcomes using different study designs and methodologies [14]. A propensity score-matched study in the US reported that after 2 years in the PCMH intervention, patients had a significant decrease in ambulatory care sensitive emergency department (ED) visits but non-significant reductions in hospital admissions and usual ED visits [15]. In contrast, declines in inpatient admissions and specialist visits were reported in an article that compared commercial health maintenance organisation (HMO) members with chronic conditions in the PCMH against non-PCMH practices in Philadelphia [16]. Additionally, the PCMH has been demonstrated to be associated with reduced healthcare utilisation and cost in patients with complex needs, including hospitalisations, ED visits, and outpatient visits [17]. On the other hand, the PCMH has been found to not have effects on the number of primary care visits, admissions and readmissions, ED visits, and diagnostic tests when the study populations included patients with and without complex needs [18,19,20].

Most studies on the integrated PCMH care model were principally carried out in the US and Australia and, of note, there is a lack of studies in Asia [8,9,10]. It is challenging to generalise findings to Asia due to different healthcare system structures and cultures, and there is a need to grow the body of literature on the PCMH in Asian contexts [21]. Furthermore, an emerging area of research is the impact of the PCMH on individuals with different comorbidity burdens. Current evidence suggests that the PCMH reduces ED use for patients with chronic conditions, including the number of avoidable and weekend ED visits, but not those without chronic conditions [20]. Patients with diabetes and hypertension experienced the greatest effects on healthcare utilisation reduction, compared to those with other chronic illnesses [20]. Older adults in the PCMH with more complex needs and at a higher risk of hospital admission had more appreciable declines in ED and inpatient service use [21]. Patients with multimorbidity and polypharmacy had lower healthcare utilisation after receiving care from the PCMH after 5 years [17]. Further research is needed to evaluate how subgroups of patients with different chronic conditions and comorbidity burdens respond to the PCMH care model to design more targeted services to address poor outcomes.

The integrated PCMH care model in this study was a component of the Community for Successful Ageing (ComSA) initiative in Singapore, an integrated care network of community-based programmes and services that delivered comprehensive and coordinated care to older adults [22,23]. The integrated PCMH model was guided by four tenets, including the life-course approach to deliver timely intervention for prevention and management of needs, ageing in place, the socioecological model of care, and population health management [22,23,24]. Our integrated PCMH intervention aimed to differentiate from existing primary care models in Singapore by targeting community-dwelling older adults with a combination of physical, psychological, and social care needs and delivering primary care and home-based care intervention elements. Current primary care models have focused on chronic disease management by multidisciplinary teams (teamlet care models) or by general practitioners at family medicine clinics [22,23].

This study aims to evaluate the effects of an integrated PCMH model of care on prospective health service use for community-dwelling older adults with complex needs. Our findings will inform practice, policy, and future research on the delivery of an integrated primary care model for ageing populations. We hypothesise that community-dwelling older adults with bio-psycho-social needs will have reductions in healthcare utilisation after receiving the integrated PCMH intervention.

## 2. Methods

### 2.1. Study Design and Study Participants

This study was part of the programme evaluation of an integrated PCMH model of care in Singapore that involved quantitative and qualitative research components [22,23,25]. We applied a quasi-experimental design for this study, whereby the intervention group consisted of study participants enrolled in the integrated PCMH and the comparator group was matched controls. We compared the difference in prospective healthcare utilisation between groups over time. Study participants in the intervention group were recruited from 1 November 2017 to 30 April 2019. Eligibility criteria were as follows: [22,23].

#### 2.1.1. Patients Aged ≥ 40 Years

A cut-off age was selected to reflect the life course approach, which applies a multidisciplinary framework to understand the role of timing in the relationships between exposures and health outcomes at the population and individual levels [22,26,27]. It conceptualises how determinants of health, experienced at different life course stages, can differentially affect health and well-being of individuals [26,27].

Additionally, the life course approach supports the delivery of healthcare interventions at an earlier life stage to be more effective at influencing the prevention and progression of chronic diseases, mental health conditions, and social care needs [26,27]. Evidence also suggests that in addition to older adults, middle-aged adults with unmet health needs have functional declines, low quality of life, high healthcare utilisation, and poor management of chronic conditions [2,28,29,30]. Hence, this study aimed to provide integrated PCMH services to adults with complex needs after their fourth decade of life to manage and prevent adverse health outcomes.

#### 2.1.2. Patients with Bio-Psycho-Social Health Risk

Defined by the 37-item Bio-Psycho-Social Risk Screener validated in the Singapore setting [30], pre-existing risk stratification criteria used by referring healthcare institutes and/or clinical assessment were included [22,23]. Functional ability and frailty were not part of the inclusion criteria.

#### 2.1.3. Patients Who Resided in the Whampoa Region

The Whampoa region is a geographically defined district in Singapore. The PCMH clinic was deliberately located in this area to serve the relatively large population of community-dwelling older persons [22].

### 2.2. Selection of Controls

Using matched controls had advantages over a single-arm before–after design by reducing bias and confounding. The control group was selected from community-dwelling older adults who resided in a neighbouring geographical location to Whampoa, the Ang Mo Kio-Cheng San (AMK-CS) region. Both regions were considered mature districts in Singapore with higher proportions of community-dwelling older adults, and persons residing in both locations had similar socio-demographic characteristics. Importantly, individuals in both regions were served by the same public regional healthcare system and had the same access to healthcare services. These included primary care, specialist outpatient, ED, and inpatient services. Only persons who resided in Whampoa could enrol in the integrated PCMH programme.

We applied coarsened exact (CE) matching to select controls. The CE matching technique may achieve better covariate balance, lower bias, and reduced model dependence, compared to other matching techniques [31,32,33]. CE matching has been recognised as a reliable strategy for selection of the comparator group for observational studies, without the need for an extensive set of baseline covariates [31,32,33]. It is ideal to be applied to studies that have a few strong confounders for matching, which is reflected in this study [34,35,36]. The CE matching process involves the coarsening of variables, which results in a larger control population that generates more accurate effect estimates with reasonable precision [34,35,36]. This technique has a range of statistical properties that are unavailable in other matching methods [31,32,33]. Furthermore, alternative methods, like propensity score (PS) matching, are being recognised as having limitations [34]. The weaknesses include sample reduction after matching and challenges in model specifications, which may unintentionally result in erroneous or misleading conclusions [34,35,36]. This study had a final analytical sample of n = 165 (shown later), so an important consideration was to prevent a potential reduction in sample size with PS matching.

Baseline covariates for matching were carefully selected and based on important variables that influenced utilisation outcomes. Baseline matching covariates included demographics (age, sex, ethnicity), socio-economic status (housing type was used as a proxy as income was unavailable), health status (weighted Charlson comorbidity index (CCI) score), and past healthcare utilisation. Matching on past healthcare use included number of primary care visits, ED visits, specialist outpatient clinic visits, and all-cause acute hospitalisations in the preceding 3 months.

There were 10,939 community-dwelling individuals available for matching. These persons were alive during the period for data analysis and had a complete dataset on baseline covariates and healthcare utilisation. Briefly, matching covariates were coarsened into substantively meaningful groups and matched controls were selected. The intervention group was categorised by calendar quarter of enrolment in the integrated PCMH. Subsequently, CEM was conducted within each category to identify controls that matched on the list of baseline covariates. The intervention group entered the PCMH programme via referrals from public primary care partners, geriatric medicine and internal medicine specialist clinics, and the public acute tertiary hospital. The controls had at least a utilisation event of a public primary care, specialist outpatient, or inpatient service in the same periods as the intervention group. Statistical weights were assigned to matched controls. A total of 5385 controls were used in the analysis.

### 2.3. Data Source

We used national administrative public healthcare records from August 2017 to 2020 to conduct CE matching and data analysis of healthcare utilisation. This period allowed us to examine changes in healthcare utilisation from the quarter (3-month period) preceding enrolment in the integrated PCMH to the five quarters post-enrolment for a follow-up of 15 months.

### 2.4. Intervention

The intervention was an integrated PCMH model of care that integrated two components, including medical care led by primary care clinicians and home-based psycho-social care led by medical social workers and nurses [22,23]. The intervention has been described in detail previously [24]. It was based on the PCMH values of patient-centredness, comprehensive and coordinated care, accessible services, shared decision making, and quality and safety [24]. One of the key objectives of the care model was to reduce the healthcare utilisation burden of patients.

The initial step of the intervention involved comprehensive bio-psycho-social assessment and the development of individualised care plans with the patient and family members [24,25]. The comprehensive assessment included medical history taking, assessment of functional needs, assessment of acute and chronic health conditions, screening for mood, cognition, and psycho-social issues, and review of medications [22,23]. Personalised care plans were delivered by a multidisciplinary team of clinicians, registered nurses, programme coordinators, and care managers. Additionally, the PCMH clinic provided linkage of patients to geriatric specialists at a tertiary acute hospital for shared care [22,23].

An aspect of the care model was the provision of home-based care management services [22,23]. Home-based care management aimed to extend the care of patients to the home setting and address the challenges faced by patients in their physical home environments [22]. Specifically, the older adults were assessed using the International Resident Assessment Instrument Home Care (interRAI-HC), which examined socio-demographics, living arrangements, cognition, communication and vision, mood and behaviour, psycho-social well-being, functional status, locomotion and walking, continence, chronic conditions, pain, nutrition, skin conditions, medications, social support networks, and depressive symptoms [22]. The social worker and nurse care managers also examined the patients’ financial and behavioural needs and provided support systems to caregivers [22,23]. The integrated PCMH ensured continuity of care and care coordination by the same care team. At subsequent PCMH clinic visits, the individualised care plans and home-based care management were reviewed [22,23].

### 2.5. Outcome Measures

The outcome measures were healthcare utilisation counts for the number of primary care visits, ED visits, specialist outpatient clinic visits, and all-cause acute hospitalisations. We examined healthcare utilisation in the five 3-month periods post-enrolment for a total follow-up of 15 months.

### 2.6. Data Analysis

We present the sample characteristics of the 165 study participants in the intervention group. Second, the profiles of the intervention and control groups after matching and the assessment of matching performance were reported. Matching performance was assessed using standardised differences, two-sample *t*-tests, and variance ratios for continuous variables. Third, we present healthcare utilisation counts per quarter (3-month period) from baseline (quarter preceding enrolment) to the five quarters post-enrolment for a follow-up of 15 months. We reported the unadjusted changes in healthcare utilisation for the intervention group for each healthcare service type at baseline and each of the five quarters post-enrolment. The descriptive statistics included mean and standard deviation (sd), median and interquartile range (IQR), and range (minimum, maximum).

Lastly, we conducted multivariable zero-inflated Poisson or zero-inflated negative binomial regression to investigate the effects of the intervention on healthcare utilisation outcome measures. Zero-inflated regression models were applied to account for the excess zero counts. Selection of the regression model was based on data distribution and meeting model assumptions. Regression adjustment of selected covariates in conjunction with matching address residual confounding and provide more robust effect estimates [34,35].

Calendar quarter was a covariate in the multivariable regression model. This study coincided with the COVID-19 pandemic period. Adjusting for calendar quarter allowed us to account for COVID-19 social distancing effects on health-seeking behaviour in the intervention and control groups. Additional covariates in the multivariable regression model were age and CCI score. The CCI score was derived from the number of chronic conditions that each person had and was assigned an integer weight from 1 to 6, with a weight of 6 representing the most severe morbidity. Summation of the weighted comorbidity scores resulted in a summary score. Weighted CCIs were based on ICD-10 codes. The list of chronic diseases included hypertension, high blood cholesterol, arthritis, eyesight problems, back pain, diabetes, hearing problems, incontinence, frequent falls, dementia, heart conditions, stroke, chronic lung disease, osteoporosis, depression, anxiety, neurological diseases, and any other condition.

Statistical significance was set at *p* < 0.05. All analyses were performed on Stata version 17.0.

## 3. Results

### 3.1. Sample Characteristics

The study participant flow diagram has been published previously [22]. From 1 November 2017 to 30 April 2019, a total of 239 older adults enrolled. Of these, 16 did not fulfil eligibility criteria. Additional older adults were excluded because they did not consent to be study participants (n = 34, 14.2%) or were uncontactable (n = 3, 1.3%). We had 184 study participants at baseline. The final sample analysed was n = 165 after loss to follow-up at 3 months (n = 11, 6.0%) and 6 months post-enrolment (n = 8, 4.3%). Six passed away, twelve withdrew as they were housebound, admitted to a long-term care facility, or no longer resided in Whampoa, and one participant was later found to be ineligible.

Table 1 shows the socio-demographics of study participants (n = 165). They had a mean age of 77.0 years and most (93.9%) were 60 years old and above. The majority were female (56.4%), married (51.5%), and Chinese (93.3%), resided in public housing (99.4%), had no formal education (48.5%), and were retired (77.0%). Weighted CCI score was 4.82.

### 3.2. Matching Performance Assessment

We assessed matching performance using standardised mean differences, two-sample *t*-tests, and variance ratios for continuous variables (Table 1). The intervention and control groups were well matched on health status, socio-economic status with housing type as its proxy, and past healthcare utilisation (i.e., number of primary care visits, ED visits, specialist outpatient clinic visits, acute hospitalisations). Both groups matched well on demographics, including sex and ethnicity. There was marginally poorer matching performance for age (intervention group: mean (sd) = 76.9 (9.9) years; controls: mean (sd) = 75.9 (9.0)).

### 3.3. Healthcare Utilisation Outcomes

Table 2 presents the descriptive changes in healthcare use for the intervention group. Appendix A shows the descriptive changes in healthcare utilisation only among those with any utilisation events.

Table 3 shows the adjusted changes in healthcare utilisation in the follow-up period, compared to matched controls.

#### 3.3.1. Number of Primary Care Visits

Compared to the mean number of primary care visits at baseline for the intervention group (mean (sd) = 0.61 (1.03)), the mean number of visits declined at every quarter (3-month period) of follow-up, with the mean number of visits being the lowest in the fifth quarter of follow-up (mean (sd) = 0.21 (0.72)) (Table 2). Based on zero-inflated multivariable Poisson regression, compared to controls, the reduction in primary care utilisation count was statistically significant in the first (β = −0.62, 95%CI = −1.00 to −0.23), second (β = −1.10, 95%CI = −1.53 to −0.68), third (β = −1.01, 95%CI = −1.49 to −0.64), and fourth (β = −1.01, 95%CI = −1.43 to −0.59) quarters of follow-up, whereby the largest reduction was in the fifth quarter of follow-up (β = −1.70, 95%CI = −2.17 to −1.22) (Table 3).

Older adults with lower comorbidity burden (CCI score < 5) had significant reductions in primary care utilisation count in the follow-up period, compared to controls (Table 3). Those with higher comorbidity burden (CCI score ≥ 5) only had significant decreases in primary care utilisation count in the first, fourth, and fifth quarters of follow-up, compared to controls.

#### 3.3.2. Number of Emergency Department Visits

Compared to the mean number of ED visits at baseline for the intervention group (mean (sd) = 0.16 (0.55)), there was a reduction in the mean number of visits in the first, second, fourth, and fifth quarters (3-month period) of follow-up. The mean number of visits was the lowest in the fifth quarter of follow-up (mean (sd) = 0.08 (0.32)) (Table 2).

Based on zero-inflated multivariable Poisson regression, the decrease in ED utilisation count was only statistically significant in the fifth quarter of follow-up (β = −0.85, 95%CI = −1.55 to −0.14), compared to the control group (Table 3). Compared to controls, older adults with lower disease burden (CCI score < 5) had a significant drop in utilisation count only in the fifth quarter of follow-up (Table 3). Compared to controls, the changes in utilisation count were non-significant in the five quarters of follow-up among those with higher comorbidity burden (CCI score ≥ 5).

#### 3.3.3. Number of Specialist Outpatient Clinic Visits

Compared to the mean number of specialist outpatient clinic visits at baseline for the intervention group (mean (sd) = 1.47 (2.18)), the mean number of visits declined in the third and fifth quarters of follow-up and it was the lowest in the fifth quarter of follow-up (mean (sd) = 0.95 (1.88)) (Table 2).

Based on zero-inflated negative binomial multivariable regression, compared to controls, the reduction in specialist outpatient clinic utilisation count was marginally out of statistical significance in the fifth quarter of follow-up (β = −0.29, 95%CI = −0.64 to 0.07) (Table 3). After stratifying by comorbidity burden, the decrease in utilisation count in the fifth quarter of follow-up was marginally outside of statistical significance for older adults with lower comorbidity burden (CCI score < 5) (Table 3). Among those with higher comorbidity burden (CCI score ≥ 5), the changes in utilisation counts were non-significant in the five quarters of follow-up, compared to controls.

#### 3.3.4. Number of All-Cause Acute Hospitalisations

The mean number of hospitalisations remained stable for the intervention group, with slight decreases in the first, second, and fifth quarters of follow-up, compared to the mean number of hospitalisations at baseline (mean (sd)= 0.11 (0.37)) (Table 2). Based on zero-inflated multivariable Poisson regression, compared to controls, the changes in hospitalisation utilisation count were non-significant and there were small effect sizes in the five quarters of follow-up (Table 3). Compared to controls, changes in utilisation count remained non-significant in the follow-up period for older adults with higher (CCI score ≥ 5) and lower (CCI score < 5) comorbidity burden.

In addition, this study examined the effects of the integrated PCMH model of care on healthcare utilisation outcomes by sex. There were no significant differences in the number of primary care visits, ED visits, SOC visits, and all-cause acute hospital admissions between males and females as a pooled sample and when stratified by comorbidity burden.

## 4. Discussion

To the best of our knowledge, this is the first study to examine the impact of an integrated PCMH on the prospective healthcare utilisation burden for community-dwelling older adults with complex needs in Singapore. It contributes to the growing evidence that integrated care encompassing comprehensive geriatric assessment, personalised care plans, multidisciplinary teams, and both in-clinic and home-based care management can have positive effects on health outcomes of older adults [6,37]. Older adults with bio-psycho-social needs in the integrated PCMH intervention had significant reductions in the use of primary care and ED services and non-significant reductions in specialist outpatient clinic services after one year, compared to community-dwelling older adult controls. Public primary care utilisation was significantly reduced from the first 3 months in the intervention and this decrease in utilisation persisted beyond one year, compared to controls. Additionally, there were no increases in hospitalisation events for older adults in the intervention group, compared to controls. Subgroup analysis demonstrated that older adults with a lower comorbidity burden may experience greater decreases in healthcare utilisation than those with higher comorbidity burden.

Comparison of findings with published literature was challenging due to the differences in the implementation of a multicomponent complex intervention in different countries, populations, and settings [14]. However, in general, our findings on the reduction in the number of primary care visits, ED visits, and specialist outpatient clinic visits appear to be largely consistent with published literature on the effects of PCMH interventions on healthcare utilisation [15,16,17,20,21,38]. This study also provides further evidence that patients with complex needs may have reduced healthcare utilisation after experiencing the integrated PCMH. This is consistent with existing evidence that a PCMH that targets chronically ill complex needs patients could better leverage on the benefits of the care model [16].

Our findings also suggest that those with more comorbidity burden, as measured by a higher CCI score, may require more interventions and time before more significant utilisation declines are seen. In the US, research has shown that for patients who were enrolled across 280 PCMH clinics, the highest decline in ED visits was among those with diabetes and hypertension [20]. Health policymakers and practitioners may need to make decisions on tailoring the PCMH interventions by comorbidity burden, be it CCI scores or specific chronic conditions.

While our study found declines in ED visits among all older adults with complex needs, a three-year prospective cohort study by Kaushal et al. (2015) on more than 200,000 patients in a PCMH programme found no significant difference in ED visits [20]. In contrast, a study on an integrated and home-based geriatric care management intervention in the US reported reduced ED visits and hospitalisation rates but only among those at a higher risk of hospitalisations [21]. The number of acute hospitalisations remained stable in this study, while there were reductions in utilisation for other service types.

We hypothesise that decreases in inpatient utilisation require more time in the programme. A study by Rosenthal et al. (2013) on a multipayer PCMH in Rhode Island also reported that downward trends in inpatient admissions only occurred after 2 years [15].

Our results on significant reductions in primary care visits within the first 3 months post-enrolment that were sustained beyond one year were likely driven by the PCMH clinic replacing the need to attend other non-PCMH primary care visits. Older patients who received integrated care by a multidisciplinary care team were reported to have lower primary care service use due to improved care coordination, continuity of care, trusting relationships with providers, and having unmet needs addressed [39,40,41,42,43].

Our findings may encourage practitioners to invest time and resources in conducting comprehensive geriatric assessment and developing care plans that account for the personalised goals and preferences of patients and caregivers. The findings also demonstrated the potential benefits of home-based care management. Existing primary care clinics could consider partnering with community-based organisations, social workers, and community nurses to extend the care of older patients to their homes.

The PCMH model has been reported to have positive effects on patient-reported outcomes and reducing cost to healthcare systems [17,22,23,42]. Hence, there is a case for healthcare policymakers to consider expanding the programme beyond pilot sites and to explore replication and scalability. We recommend healthcare systems to review the scaling up of integrated care models, including the PCMH model, to expand the intervention to a greater proportion of older adults with complex needs. Further research could examine the outcomes of additional healthcare service types and for a longer follow-up period to provide a more comprehensive understanding of whether decreases in healthcare utilisation can be sustained and which services require more time for significant reductions in utilisation.

Another point is that our subgroup analysis by comorbidity burden suggested that better segmentation of older patients is required in the community. Those with greater comorbidity burden have higher mortality and medical needs. A more intensive approach in the community to prevent acute care utilisation is needed. Further work is recommended to examine how the current model of PCMH can be changed or improved to adequately manage these patients.

Studies on the barriers to scale will inform healthcare policymakers on the decisions and resources needed for successful expansion of the PCMH model. A second consideration is expansion to other groups. Participants in this study were older adults with a mean age of 77 years. Future studies could compare the effects of an integrated PCMH care model on older persons of lower age groups.

Lastly, qualitative studies play an important role in eliciting the mechanisms of change in how an integrated PCMH model reduces healthcare utilisation [41]. Our recent publication reported that patients benefited from shared decision making, provider–patient relationships, and engagement of family members and caregivers [25]. Anastas et al. (2019) conducted focus group discussions with patients and implementers and found that continuity of care and adoption of care plans were among the high-value elements that could enable healthcare use declines [40].

Further research is recommended. For instance, programme implementers and healthcare staff could provide their perspectives on how elements of the PCMH intervention enabled the reductions in healthcare service use or how utilisation declines can be sustained. Another recommendation for future research is to examine how the impacts of the integrated PCMH care model on healthcare utilisation and other health outcomes are influenced by socio-demographics like sex and highest educational attainment, as well as cognitive function.

The literature suggests that there are differences in healthcare utilisation in older persons by sex and education [43,44]. However, it is still not well understood how the differences are driven by these demographic factors. It has been suggested that the differences in healthcare use and other health outcomes between females and males could be driven by factors like financial access and social isolation, instead of sex per se [43,44,45]. In addition, the prevalence of cognitive decline in older adults is expected to increase over the next few years [46]. Future studies could investigate the changes in healthcare utilisation after receiving the integrated PCMH care intervention by subgroups of older adults with different cognitive function. We also recommend that future studies include screening for cognitive function in both the intervention and control groups to allow for subgroup analysis by cognition. Additionally, the literature has proposed the screening of cognitive function, among other health domains, within integrated care models [46]. These findings will inform policy and practice on, firstly, which subgroups benefit the most from the integrated PCMH care interventions and, secondly, how the integrated PCMH model of care could potentially be tailored for different subgroups of patients.

### Strengths and Limitations

First, a key strength of this study was that we used national administrative data to analyse healthcare utilisation of outpatient and inpatient services. This methodology has advantages over self-reported healthcare utilisation, which may be subject to reporting bias and recall error. Second, we applied a quasi-experimental approach by comparing healthcare utilisation over time with well-matched controls. Matching covariates were carefully selected to ensure controls were similar in variables that influenced outcomes like socio-demographics, health status, and past utilisation. Third, we evaluated a relatively novel model of care that integrated primary care and homecare to meet the multidimensional needs of older patients. This article addresses an important research question on how an integrated PCMH model that has aspects of comprehensive assessment in the home and clinic settings, a multidisciplinary team, personalised care plans, and shared decision making influences the burden of healthcare service use on patients. Fourth, the study had a follow-up period of 15 months, which allowed us to elicit insights into utilisation patterns beyond one year.

One limitation was the lack of administrative data on other types of healthcare services, including day surgeries, day rehabilitation services, dental services, and other allied health services. Second, our study found that the number of hospitalisations did not change for the intervention group, compared to controls. We did not have data on cumulative length of stay to examine impact on the duration of hospitalisations. Additionally, we did not have administrative data on the fees or patient payable amounts for hospital stays. Due to the limitation of the available administrative data and inability to derive accurate estimates of costs of hospital stays, this study did not report hospital fees paid by patients. This study still contributes to the literature by providing evidence on the impact of an integrated PCMH care model on healthcare utilisation for older adults with complex bio-psycho-social needs and the effects of differential comorbidity burden. Third, while our study had a relatively long follow-up, more continued follow-up is needed to ascertain whether reductions in healthcare use were sustained and whether declines in hospitalisation utilisation count would require older persons to be in the programme for a longer time [15,21]. Lastly, the study design is observational for pragmatic reasons. A randomised controlled trial (RCT) would reduce confounding by observable and unobservable factors but would not be practical or feasible for a complex intervention with multiple care components. Having a control study site that delivered usual care may have led to higher refusal rates and lower recruitment. Importantly, there were ethical considerations for not providing older adults with complex needs with comprehensive geriatric assessment and individualised care management. Finally, we recognise that more statistically significant findings could have been elicited if we had a larger sample size.

## 5. Conclusions

Our previous publications have shown that the intervention had positive effects on needs satisfaction, patient activation, and patient experience [23,25]. This study shows that an integrated PCMH was associated with reduced healthcare utilisation and also decreased cost to the healthcare system [22,23,25]. Together, these findings suggest that the PCMH model of care would likely be beneficial to community-dwelling older adults with complex needs. Future research could evaluate the scalability and sustainability of the care model, its impact on long-term health service use, and how to tailor the interventions to various subgroups of individuals such as by age and comorbidity burden.

## Figures and Tables

**Table 1 ijerph-20-06848-t001:** Post-coarsened exact matching: Profile of intervention group (n = 165) and control group (n = 5385).

Variable	Intervention Group (PCMH Patients) (n = 165)	Coarsened Exact-Matched Control Group (n = 5385)	Standardised Difference	*t*-Test (Two-Sample *t*-Test)	Variance Ratio for Continuous Variables (Treat vs. Control)
**Socio-demographics**					
**Age, years**, mean (sd)	76.9 (9.9)	75.9 (9.0)	**0.11**	*p*-value > 0.05	1.21
**Sex**, **%**					Not applicable
Male	72 (43.6%)	2348 (43.6%)	0.00		
Female	93 (55.4%)	2983 (55.4%)	0.00		
**Ethnicity, %**					
Chinese	154 (93.3%)	5026 (93.3%)	0.00		
Malay	3 (1.8%)	98 (1.8%)	0.00		
Indian	7 (4.2%)	228 (4.2%)	0.00		
Others	1 (0.6%)	33 (0.6%)			
**Housing type, %**			0.00		
Public housing	164 (99.4%)	5353 (99.4%)	0.00		
Condominium/Landed property	1 (0.6%)	32 (0.6%)	0.00		
**Health status**					
**CCI score ^a^**, mean (sd)	4.82 (2.01)	4.53 (2.83)	0.01	*p*-value > 0.05	0.50
**Past healthcare utilisation: Healthcare utilisation in the 3-month period (one quarter) pre-enrolment**					
No. hospital admissions, mean (sd)	0.11 (0.37)	0.14 (0.33)	0.09	*p*-value > 0.05	1.26
No. ED visits, mean (sd)	0.16 (0.55)	0.18 (0.47)	0.04		1.37
No. SOC visits, mean (sd)	1.47 (2.18)	1.32 (1.88)	0.07		1.34
No. public primary care visits, mean (sd)	0.61 (1.03)	0.68 (1.06)	0.07		0.61
**Education, n (%)**					
No formal education	80 (48.5%)	Not applicable (No data available for non-PCMH controls)			
Primary school	51 (30.9%)				
Secondary school	23 (13.9%)				
Post-secondary (non-tertiary)	8 (4.8%)				
Diploma and professional	3 (1.8%)				
**Employment status, n (%)**					
Employed full-time	14 (8.5%)	Not applicable (No data available for non-PCMH controls)			
Employed part-time	13 (7.9%)				
Unemployed	7 (4.2%)				
Retired	127 (77.0%)				
Others	4 (2.4%)				

Bold: Indicator of poor matching performance: SMD ≥ 0.1, *t*-statistic *p*-value < 0.05, variance ratio < 0.5 or >2.0; CCI: Charlson comorbidity index; ^a^ The CCI score was derived from the number of chronic conditions that each person had and was assigned an integer weight from 1 to 6, with a weight of 6 representing the most severe morbidity. Summation of the weighted comorbidity scores resulted in a summary score. Weighted CCIs were based on ICD-10 codes. The list of chronic diseases included hypertension, high blood cholesterol, arthritis, eyesight problems, back pain, diabetes, hearing problems, incontinence, frequent falls, dementia, heart conditions, stroke, chronic lung disease, osteoporosis, depression, anxiety, neurological diseases, and others.

**Table 2 ijerph-20-06848-t002:** Descriptive analysis for healthcare utilisation (intervention group n = 165).

Utilisation for Each Health Service	Pre-Enrolment	Follow-Up Period Post-Enrolment				
	One Quarter before Enrolment	1st Quarter	2nd Quarter	3rd Quarter	4th Quarter	5th Quarter
**Number of public primary care visits**						
**Pooled (n = 165)**						
Mean (sd)	0.61 (1.03)	0.33 (0.77)	0.22 (0.57)	0.22 (0.64)	0.24 (0.72)	0.21 (0.72)
Median (IQR)	0 (1–2)	0 (0–0)	0 (0–0)	0 (0–0)	0 (0–0)	0 (0–0)
Range (min–max)	0–5	0–3	0–3	0–4	0–5	0–6
**Number of emergency department visits**						
**Pooled (n = 165)**						
Mean (sd)	0.16 (0.55)	0.15 (0.41)	0.15 (0.39)	0.19 (0.50)	0.12 (0.39)	0.08 (0.32)
Median (IQR)	0 (0–0)	0 (0–0)	0 (0–0)	0 (0–0)	0 (0–0)	0 (0–0)
Range (min–max)	0–4	0–2	0–2	0–2	0–3	0–2
**Number of specialist outpatient clinic visits**						
**Pooled (n = 165)**						
Mean (sd)	1.47 (2.18)	1.56 (2.72)	1.96 (4.46)	1.36 (2.34)	1.29 (2.33)	0.95 (1.88)
Median (IQR)	0 (0–2)	0 (0–3)	0 (0–2)	0 (0–2)	0 (0–2)	0 (0–1)
Range (min–max)	0–10	0–18	0–44	0–14	0–14	0–13
**Number of all-cause acute hospitalisations**						
**Pooled (n = 165)**						
Mean (sd)	0.11 (0.37)	0.10 (0.30)	0.10 (0.30)	0.12 (0.39)	0.11 (0.38)	0.10 (0.36)
Median (IQR)	0 (0–0)	0 (0–0)	0 (0–0)	0 (0–0)	0 (0–0)	0 (0–0)
Range (min–max)	0–2	0–1	0–1	0–2	0–3	0–2

Quarter: Refers to a 3-month period, e.g., 4th quarter refers to the fourth 3-month period post-enrolment date in PCMH for the intervention group.

**Table 3 ijerph-20-06848-t003:** Adjusted changes in healthcare utilisation in the follow-up period, compared to controls (n = 165).

Healthcare Service	Follow-Up Period Post-Enrolment				
	1st Quarter (β, 95% CI)	2nd Quarter (β, 95% CI)	3rd Quarter (β, 95% CI)	4th Quarter (β, 95% CI)	5th Quarter (β, 95% CI)
**No. of public primary care visits ***					
Pooled (n = 165)	**−0.62** **(−1.00, −0.23)**	**−1.10** **(−1.53, −0.68)**	**−1.06** **(−1.49, −0.64)**	**−1.01** **(−1.43, −0.59)**	**−1.70** **(−2.17, −1.22)**
CCI score ≥ 5 (n = 54)	**−1.78** **(−3.38, −0.19)**	−16.66 (−24.21, 23.89)	−1.59 (−3.19, 0.01)	**−2.30** **(−4.41, −0.20)**	**−2.06** **(−3.62, −0.49)**
CCI score < 5 (n = 111)	**−0.54** **(−0.94, −0.14)**	**−1.00** **(−1.44, −0.56)**	**−1.01** **(−1.45, −0.56)**	**−0.92** **(−1.36, −0.49)**	**−1.64** **(−2.14, −1.15)**
**No. of emergency department visits ***					
Pooled (n = 165)	−0.17 (−0.79, 0.45)	−0.10 (−0.74, 0.54)	0.09 (−0.51, 0.69)	−0.15 (−0.84, 0.54)	**−0.85** **(−1.55, −0.14)**
CCI score ≥ 5 (n = 54)	0.31 (−1.63, 2.25)	0.10 (−2.07, 2.28)	0.94 (−0.88, 2.76)	0.07 (−2.11, 2.25)	−1.04 (−3.60, 1.53)
CCI score < 5 (n = 111)	−0.2 (−0.87, 0.46)	−0.14 (−0.81, 0.53)	−0.04 (−0.69, 0.60)	−0.11 (−0.86, 0.65)	**−0.88** **(−1.82, −0.15)**
**No. of specialist outpatient clinic visits ****					
Pooled (n = 165)	0.09 (−0.27, 0.45)	0.37 (0.01, 0.73)	−0.04 (−0.40, 0.33)	0.06 (−0.32, 0.44)	**−0.29** (−0.64, 0.07) ^^^
CCI score ≥ 5 (n = 54)	−0.15 (−1.30, 1.01)	0.54 (−0.65, 1.73)	0.34 (−0.86, 1.54)	0.19 (−0.98, 1.36)	0.18 (−0.91,1.27)
CCI score < 5 (n = 111)	0.11 (−0.27, 0.48)	0.35 (−0.02, 0.72)	−0.07 (−0.45, 0.31)	0.05 (−0.34, 0.45)	**−0.33** (−0.70, 0.05) ^^^
**No. of all-cause acute hospitalisations ***					
Pooled (n = 165)	−0.22 (−0.97, 0.53)	−0.13 (−0.88, 0.62)	0.02 (−0.72, 0.75)	0.10 (−0.67, 0.86)	−0.56 (−1.32, 0.21)
CCI score ≥ 5 (n = 54)	0.89 (−1.54, 3.31)	0.65 (−1.96, 3.26)	0.63 (−2.00, 3.26)	0.58 (−2.03, 3.20)	−14.34 (−20.15, 19.86)
CCI score < 5 (n = 111)	−0.19 (−1.03, 0.64)	−0.16 (−0.96, 0.64)	−0.04 (−0.81, 0.74)	0.26 (−0.59, 1.10)	−0.57 (−1.34, 0.20)

Reference category: one quarter (3-months) before enrolment; Quarter: Refers to a 3-month period, e.g., 4th quarter refers to the fourth 3-month period post-enrolment date in PCMH for the intervention group. ***** Multivariable zero-inflated Poisson regression adjusted for calendar time, age, and CCI. ****** Multivariable zero-inflated negative binomial regression adjusted for calendar time, age, and CCI. Bold: *p*-value < 0.05. **^** Marginally out of statistical significance.

## Data Availability

There are no unpublished data available. The corresponding author can be contacted on this matter.

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
