# Peer review of "An Integrated Patient-Centred Medical Home (PCMH) Care Model Reduces Prospective Healthcare Utilisation for Community-Dwelling Older Adults with Complex Needs: A Matched Observational Study in Singapore"

_ijerph, 2023, doi:10.3390/ijerph20196848_

Round 1
Reviewer 1 Report
I would like to congratulate the authors because it seems to me a very relevant topic.
1) In the section “Introduction”, I don't think this sentence is appropriate “Existing studies on the PCMH are also disproportionately in the US and Australia.10- 66”, in my opinion never studying a subject is disproportionate; could you modify it?
2) In the section “Methods”; could the authors further explain why they made the age cutoff at patients equal to or older than 40 years of age?
3) Have the authors considered whether there would be any differences with respect to this model of care based on the gender of the patients? I recommend analyzing this and seeing if there are any differences.
4) Do the authors consider that this type of program could be influenced by the educational level and cognitive level of older adults? Has this been contemplated in any part of the study? Unfortunately, as we advance in the life cycle, different types of cognitive dysfunctions could appear and it would be of great interest to be able to contemplate how they could be approached in this type of programs.
Reviewer 2 Report
I suggest that the researchers should collect data about fees for every hospital visit as you intend to test the effects to reduce patients' burdens, but the numbers of hospital visits cannot be used to clarify this.
Round 2
Reviewer 2 Report
No suggestions.
Author Response
Thank you for reviewing our manuscript.